# High Methoxyl Pectin and Sodium Caseinate Film Matrix Reinforced with Green Carbon Quantum Dots: Rheological and Mechanical Studies

**DOI:** 10.3390/membranes12070695

**Published:** 2022-07-07

**Authors:** Clarissa Murru, Mohammad Amin Mohammadifar, Jakob Birkedal Wagner, Rosana Badía Laiño, Marta Elena Díaz García

**Affiliations:** 1Department of Physical and Analytical Chemistry, Faculty of Chemistry, University of Oviedo, 33006 Asturias, Spain; rbadia@uniovi.es (R.B.L.); medg@uniovi.es (M.E.D.G.); 2Research Group for Food Production Engineering, National Food Institute, Technical University of Denmark, Søltofts Plads, 2800 Kongens Lyngby, Denmark; moamo@food.dtu.dk; 3DTU Nanolab, Technical University of Denmark, Fysikvej, 2800 Kongens Lyngby, Denmark; jabw@dtu.dk

**Keywords:** composite films, nano additives, rheology, mechanical properties

## Abstract

Nowadays, proteins and polysaccharides play a fundamental role in the manufacturing of biocompatible materials applied in food packaging. The resulting films have, however, limits associated with the resistance to mechanical stress; therefore, it is important to reinforce the initial mixture with additives that promote the development of stronger molecular links. Carbon dots (CDs) are excellent candidates for this purpose due to the presence of surface functional groups that determine the formation of numerous intramolecular bonds between the charged biopolymers. The present research aims to evaluate the effect of CDs on the mechanical properties of biopolymer films obtained from sodium caseinate (CAS), high methoxyl pectin (HMP) and glycerol used as plasticizers. Green carbon dots (gCDs) were obtained from natural organic sources by green synthesis. The effects of gCDs on the flow behavior and viscoelastic properties of mixed biopolymer dispersions and the thermophysical properties of the corresponded films were evaluated by steady and unsteady shear rheological measurements and differential scanning calorimetry (DSC) tests, respectively. The dynamic mechanical measurements were realized taking into account the parameters of temperature and relative humidity. The results indicate a significant change in the viscosity of the protein–polysaccharide dispersions and the thermomechanical properties of the corresponding film samples reinforced with higher amounts of gCDs.

## 1. Introduction

Due to the increasing awareness of the population regarding the environmental impact and the sustainability of plastic packaging, new biomaterials that can replace single-use plastic are being developed. Consumers demand high-quality and safer foods, but also packages that do not increase pollution and are made by sustainable processes. The potential advantages of biobased and biodegradable plastics, in comparison to synthetic plastics, include: (i) lower carbon footprint, (ii) preparation from renewable resources, (iii) less energy consumption during synthesis and (iv) fewer greenhouse gases. Moreover, the possibility to add agro-wastes as blends for preparing biocomposites is an innovative approach that allows for the improvement of their compostability and accelerate the degradation processes [1]. Bioplastics can derive from polymers that are both bio-based and biodegradable, such as polylactic acid (PLA) and polyhydroxyalkanoates (PHAs), as well as plastics based on starch, cellulose, polysaccharides and proteins. However, they can also derive from polymers that are only bio-based but not biodegradable (bio-based polyamides, polyethylene and polyethylene terephthalate) and those that are only biodegradable (polycaprolactone, polyvinyl alcohol and polybutylene adipate terephthalate) [2]. The films and coatings used for food packaging are usually obtained from polysaccharides, proteins and lipids, though the products resulting from these materials have limitations. Despite polysaccharide-based films providing good barrier properties against oils and lipids, their efficiency decreases against moisture while films obtained from proteins are characterized by their poor mechanical strength [3]. The research for innovative materials has recently focused on the fabrication of multicomponent films to explore the mandatory advantages of each constituent, as well as to minimize their disadvantages [4]. Blends of protein and polysaccharide in specific solution conditions (protein/polysaccharide ratio, ionic strength and pH of solution) can result in protein–polysaccharide complexes that form films with advantageous physical properties. The main driving force involved in the complexation of proteins and polysaccharides is electrostatic interactions that can lead to the formation of soluble and insoluble complexes depending on the equality of net charge on the complex. The functional properties and resistance of the resulting films might be improved by cross-linking proteins and polysaccharides, or by applying treatments and specific additives contributing to the establishment of interactions between the biopolymers [5]. Various nanofillers have been used to prepare biopolymer-based nanocomposite films and, recently, carbon dots (CDs) have emerged as potential materials for food packaging applications [6]. These nanoparticles present valuable and remarkable properties, such as low toxicity, water solubility, high biocompatibility, photochemical and physic-chemical stability, low production costs and photoluminescence. All of these characteristics make them very attractive in a whole gamma of applications, such as bioimaging, drug administration, biodetection, disease diagnosis, synthetic chemistry and material science [7]. The most common pathways of synthesis of CDs are the hydrothermal or solvothermal ones, due to their simplicity of operation and the low production cost. Furthermore, the possibility of using natural sources as well as organic subproducts or wastes as the initial carbon source for the synthesis encompasses the CDs in the green chemistry and circular economy frames [8,9]. However, it is the highly multifunctionalized surface that CDs possess that has awakened the interest in the field of nanocomposite development. Bare CDs are commonly rich in epoxy and carboxylic groups, although they might contain other functionalities, such as –SH_2_, -NH_2_ and alkyls, depending on the original materials used as a carbon source. Thus, either with the multifunctionalities naturally present or with further derivatizations using those as linkage points, a tailor-functionalized CDs can be obtained without difficulty [7,9]. In this work, we fabricate bio-films starting from Na-caseinate (CAS) and highly methoxylated pectin (HMP) using glycerol as a plasticizer to study the rheological and mechanical influences of CDs added to the mixture as enhancers. CAS was selected due to its unique properties, such as water solubility, capability to form films, high oxygen barrier properties (higher than non-ionic polysaccharides) and its high cohesive energy density [10]. Moreover, it has the ability to form a variety of interactions with other molecules (hydrogen and ionic bonds, van der Waals, hydrophobic and electrostatic interactions), which makes it a good candidate to produce bio-compatible films with good resistance [11]. On the other hand, highly methoxylated pectin (HMP) is an anionic polysaccharide that has been widely used in the pharmaceutical and food industries due to its gelling properties [12]. The chemical characteristics of pectin make it an ideal material for the improvement of the mechanical properties of films obtained from sodium caseinate with a strong, but not very elastic, structure. The CDs were obtained by hydrothermal synthesis from apple pomace (APCDs) and rosemary leaves (RCDs), and they were selected in this work for their interesting characteristics, such as renewable sources and/or waste products from food industries and their potential low cytotoxicity [13,14]. The applications of CDs as enhancers of the mechanical properties of films is still in its infancy. The purpose of this work is to investigate the influence of green carbon-based nanomaterials in the development of a stronger molecular network in the CAS/HMP matrix and to explore the effects of temperature and humidity in the response to the mechanical tests conducted on the films obtained. The novel outcome of this study may reveal a new insight into the exploitation of CDs as fillers in biocompatible films with possible applications in food packaging.

## 2. Materials and Methods

### 2.1. Materials and Chemicals

Rosemary powder was obtained from a Danish local supermarket (Netto, Regensburg, Germany), and depectinized apple pomace, Herbavital F12, was supplied by Herbstreith & Fox GmbH & Co. KG Pektin-Fabriken (Neuenbürg, Germany). Sodium caseinate powder (CAS), methanol, ethanol and chloroform for HPLC were supplied by Merck Co. (Darmstadt, Germany). High methoxyl pectin powder (HMP) GENU^®^ Pectin, 50–75% esterification, was purchased from CPKelco (Lille Skensved, Denmark), and Glycerol 99% (PlusOne Glycerol) was from GE Healthcare (Chicago, IL, USA). All solutions were prepared using distilled water.

### 2.2. Instrumentation

Images of the casein/pectin film structure were obtained through an inverted microscope (Olympus XI 51), while surface images were obtained with a scanning electron microscope (AFEG 250 Analytical ESEM). Following the synthesis, the gCDs were purified using the VibroTM-LE System (SANI Membranes, Farum, Denmark) with a membrane of 5 kDa MWCO. The morphological details and size determinations of the gCDs were attained from the images obtained using a high-resolution transmission electron microscope (TEM JEOL-JEM 2100F). Rheological and mechanical studies of casein, pectin and carbon dot suspensions were performed by using a rheometer (TA Instruments, New Castel, DE, USA) coupled with a concentric cylinder. Measurements of the physical properties at controlled temperature and relative humidity of biopolymer films were obtained using the above described rheometer coupled with a film tension accessory combined with an environmental chamber. The thermal behavior of the gCDs and biopolymer films was obtained using a differential scanning calorimetry (DSC 250, TA Instruments, Newcastle, DE, USA) equipped with Trios software and a Refrigerated Cooling System 90. These experiments were conducted using approximately 5 mg of the samples weighed into an aluminum container that was subsequently hermetically closed with a lid. An empty container with the same characteristics was used as a reference.

#### 2.2.1. Rheological Characterization

Oscillatory shear tests were performed on a Discovery Hybrid Rheometer DHR-2 (TA Instruments, Newcastle, DE, USA) using a concentric cylinder. Temperature control was performed with a Peltier system equipped with a fluid circulator. The strain sweep tests were performed to determine the linear viscoelastic region (LVR), in which the properties of the materials, such as the structure, were unaffected by the applied stress [15]. Frequency sweeps were made over the frequency range from 0.1 to 10 Hz and a low-amplitude oscillatory shear of 1%. Samples were individually loaded in the cylinder and allowed to stand at 25 °C for 5 min before the test. All tests were performed in three replicates, each of which was analyzed once.

#### 2.2.2. Mechanical Characterization

Dynamic mechanical tests (DMAs) were performed using a rheometer coupled with a Film/Fiber Tension geometry in an environmental chamber (TA Instruments, Newcastle, DE, USA). This innovative analysis technique allows for the study of biopolymer films under different conditions of relative humidity and temperature. To proceed with the measurements, rectangular pieces were cut from the biopolymer films (approximately 3 × 1 cm^2^) and placed inside the chamber. The thickness, length and width of each piece were measured using a Vernier caliper, and the values obtained were used to set the instrument before analysis. An axial force of 0.01% and a frequency of 1 Hz were established for the realization of the tests. The starting temperature used in the heating ramp was 25 °C and the final temperature was 45 °C, with a temperature increase of 0.7 °C/min.

### 2.3. Preparation of the Sodium Caseinate and Pectin Suspensions

Two stock suspensions of CAS and HMP were prepared in distilled water at a concentration of 2% (*w*/*w*), with gentle stirring for 4 h at room temperature; the final solutions were then stored at 5 °C for 18 h to ensure the complete hydration of the compounds. The suspensions used for the study were prepared by adding dropwise an appropriate amount of CAS suspension to a given volume of HMP suspension at room temperature under continuous stirring. The resulting mixture was finally stirred for 90 min to homogenize it. The suspension for the preparation of the films was obtained by mixing sodium caseinate (CAS) and highly methoxylated pectin (HMP) in a 1:3 ratio. The choice of this combination was based on the previous study conducted by M. Jahromi et al. [16], which offered good results in terms of the resistance and elasticity of the film structure.

### 2.4. Synthesis of gCDs from Apple Pomace and Rosemary Leaves

The gCDs were obtained from rosemary and apple pomace using a hydrothermal synthesis according to the following protocol: 20 g of each raw material was placed in separate Pyrex glass beakers and heated in an oven for 5 h at 200 °C. During the heating process, 5 mL of distilled water was added to prevent the product from scorching every 30 min. Once the beakers were cooled to room temperature, the black powder obtained was mixed with 20 mL of distilled water and centrifuged at 12,000 rpm for 50 min to remove any precipitate. The supernatants were vacuum filtered through a 0.40 µm pore paper filter, and then purified using a continuous nanofiltration system (MWCO 5 kDa) for 1 h. Finally, the concentrated solutions were lyophilized and the CDs obtained were stored at 2 °C until later use. The synthesis yield was found to be 5.8% *w*/*w* for rosemary CDs (RCDs) and 4.5% *w*/*w* for apple pomace CDs (APCDs). All the CDs were found to be water and ethanol dispersible.

#### Preparation of CAS/HMP Suspensions and Biopolymer Films Doped with gCDs

For the preparation of CAS/HMP suspensions enriched with different amounts of gCDs, 25 mL of CAS 2% *v*/*v* and 75 mL of HMP 2% *v*/*v* were mixed, and then the necessary volume of CDs solutions was added to reach concentrations of 0%, 0.25%, 0.5% and 1.0% *w*/*w*. In the case of biopolymer film preparations, 40 mL of each of the above suspensions containing glycerol (0.75% *w*/*w*) were poured into plastic Petri dishes (90 mm diameter) and dried for 24 h at 40 °C. Then, dried biopolymeric films were peeled off and conditioned for 72 h at room temperature in a desiccator containing a saturated magnesium nitrate solution to maintain a constant 53% relative humidity.

## 3. Results and Discussion

### 3.1. The Characterization of Green Carbon Dots 

High-resolution transmission electron microscopy (HRTEM) images of the synthesized gCDs demonstrated that both types of nanoparticles presented roughly spherical shapes (Figure 1) with a mostly monodispersed size distribution. A nanoparticle average diameter of 3.8 ± 0.7 nm was found for RCDs (Figure 1a), while the magnified HRTEM image showed amorphous and crystalline phases with an interplanar spacing of 0.21 nm, which corresponded to the interplane spacing of the graphite (d100 = 0.213 nm) [17]. The APCDs (Figure 1b) were slightly larger with an average diameter of 4.2 ± 0.9 nm.

#### 3.1.1. Morphological Characterization of the CAS/HMP Suspensions

Several CAS/HMP precursor suspensions were prepared with increasing amounts of pectin, with CAS:HMP ratios ranging from 100:0 to 0:100 *v*/*v* with pH values of 7.15 ± 0.2 and 3.34 ± 0.3, respectively, at room temperature. During the whole duration of the experiments, the suspensions did not show any flaculations or the formation of a second phase, and the structure of the obtained films was finally observed using an inverted microscope. The amphiphilic nature of casein makes possible the self-assembly into stable micellar structures when dispersed in aqueous solutions. The characteristic “spongy” structure is visible in the sample that presents the highest concentration of Na-caseinate, with a CAS:HMP ratio of 75:25 *v*/*v* (Figure 2a). Due to the increase in the quantity of pectin and the consequent acidification of the medium, the micelles presented a net positive charge, while the negatively charged pectin was assumed to bind to the casein micelles via electrostatic forces. Therefore, it was possible to observe that the composition of CAS:HMP of 50:50 resulted in an assemblage of uniformly distributed micelles of smaller sizes (Figure 2b) due to the limited aggregation process in the presence of higher HMP concentrations. At a CAS:HMP composition ratio of 25:75, the characteristic micellar structure was no longer visible. This could have been caused by two different reasons: the low Na-caseinate concentration was not enough to reach the stoichiometric point allowing the self-assembly into micelles; and/or, as described by Wusigale et al. [18], the gradual increase in the pectin concentration results in a full coverage of the casein micelles by a pectin layer lowered the attraction among the particles (Figure 2c). In Figure 2d, it is possible to observe a schematization of the rearrangement of the structure of casein and pectin based on the increase in the concentration of HMP and the consequent acidification of the medium.

#### 3.1.2. Steady-State Shear Rate

In this section, the effects of the shear rate on the viscosity of neat polymers and gCDs polymer matrices were studied. The results plotted in Figure 3 indicate that the addition of gCDs to the CAS/HMP suspensions lead to an increase in viscosity in the systems. Non-Newtonian behavior was observed for all the samples, being more pronounced for CAS/HMP + 0.5% RCDs and CAS–HMP + 1.0% RCDs than for the neat CAS/HMP and CAS/HMP + 0.25% RCDs. The behavior of the samples can be explained by the molecular interactions between the polymer chains and the gCDs present in the dispersion: in the neat CAS–HMP matrix, the random orientation of casein and pectin chains is rapidly aligned in the direction of the flow as the shear rate increases, resulting in poor interactions among the chains and lower viscosity values at higher shear rates. However, the addition of gCDs to the medium can promote the formation of agglomerates due to the morphology of the nanoparticles [19] and/or to the solvent in which the compounds are suspended [20], leading to an increase in the viscosity of the sample. Similar results were observed by Mohd et al. in polyacrylamide suspensions enriched with nanoparticles [21]. To better understand the behavior of each sample, the obtained data were fitted to the power-law (Ostwald–de Waele) model [22]:
(1)$$\eta ={K\;\dot{\gamma }}^{\;n-1}$$
where η (Pa.s) is the viscosity, K (Pa.sn) is the consistency factor, $\dot{\gamma }$
^(n − 1)^ is the shear rate and n is the flow behavior index. Depending on the value of n, the fluid can be classified as being pseudoplastic (n < 1), Newtonian (n = 1) or dilatant (n > 1). In Figure 3, the resultant curves indicate that the studied mixtures have a shear-thinning tendency and comparable viscosity profiles. The samples containing a higher percentage of RCDs exhibited the greatest viscosity during the shear-rate test interval, while blends containing only 0.25% RCDs and CAS/HMP (i.e., control) exhibited the lowest. According to the references [23], with the addition of the RCDs to the polymers matrix, new networks of polymers and gCDs could be created. These additional networks affected the behavior of the blends, impeding the movement of polymer chains and increasing the viscosity of the mixtures. A 1% RCD sample shows the lowest n value and therefore the highest pseudoplastic tendency, indicating that the concentration of gCDs plays a fundamental role in the formation of stronger molecular interactions as was also observed by R. Kotsilkova et al. [24]. The values of K and n are collected in the table inserted in Figure 3. It is possible to observe that, although all the samples present values of n < 1, indicating a deviation from the Newtonian behavior, the K factor and n index of CAS/HMP and 0.25% RCDs are almost identical, demonstrating that the addition of a minimum concentration of 0.5% w/w of gCDs to the medium is necessary to promote the formation of stronger interactions that increase the viscosity. Comparable results were collected by Arrigo and Malucelli [25]. On the basis of the results, we can conclude that the addition of a certain concentration of nanoparticles can improve the pseudo-plasticity behavior of a CAS/HMP polymer solution at a given shear rate. The results show a similar trend in the samples obtained using the gCDs derived from apple pomace (Appendix A).

### 3.2. Thermal Behavior

The thermal properties of materials are investigated when they are designed for industrial applications. Differential scanning calorimetry (DSC) is a tool that provides valuable information about the changes in the glass transition temperature and thermal degradation of materials [26]. The glass transition temperature (Tg) is the temperature below which polymer molecules remain in a glassy state and above which they reach a viscous liquid state. The DSC measurements obtained in this study derive from a single scan where the temperature was increased from 175 to 425 K during a total time of 80 min. DSC results of CAS/HMP and CAS/HMP-RCDs membranes, as well as the neat RCDs, are presented in Figure 4. Tg around 257 K was observed for the CAS/HMP membrane, which was much lower than that reported by Jahromi et al. [16] (270 K) for a similar CAS/HMP composition. It is also interesting to notice that, as the concentration of added RCDs increased, a monotonical decrease in the Tg was observed: 251 K for 0.25% RCDs, 246 K for 0.50% RCDs and 245 for 1.0% RCDs. The presence of RCDs may limit the close packing of the CAS and HMP polymer chains, resulting in a decrease in Tg as the RCDs concentration increases. Some groups reported a decrease in Tg in different polymer nanocomposites and explanations are varying. Serenko et al. [27] described that, through a thermodynamic analysis, the increased surface of small-sized nanoparticles (1–3 nm) can increment the conformational states associated with different orientations of the polymer macromolecules and the nanoparticles themselves. Similar behavior was shown in presence of APCDs (Appendix A).

### 3.3. Rheological Characterization of CAS/HMP Suspensions Enriched with gCDs

The rheological studies were realized through an oscillatory test, a fundamental tool which allows for the study of the energy storage and dissipation modules and the determination of the viscoelastic properties of fluids. The storage modulus G’ is a measure of the magnitude of energy stored in the material (or recoverable per cycle of deformation), while the loss modulus G′′ represents the energy that is lost at viscous dissipation per cycle of deformation. Therefore, for a perfectly elastic material, all the energy is stored (G′′ equals zero), while, for a material with no elastic properties, all the energy is dissipated as heat (G′ equals zero). Other important functions for the description of viscoelastic behavior are the complex modulus G*, which is defined as the square root of the complex combination of elastic and loss modulus and the loss tangent (or tangent of the phase shift), tan δ, which is the ratio of viscose to elastic components, tan δ = G′′/G′. The phase angle δ varies between 0 and 90°: it is zero for a purely elastic material and 90° for a purely viscous fluid [28]. The frequency dependence of G* could be used to evaluate the type of structure of fluids: samples with a less developed structure show more frequency dependency. The slope of the log plot of complex modulus versus frequency for neat CAS/HMP suspensions and those containing different amounts of RCDs (Figure 5 and Appendix A for APCDs) approached 1,1, which means that a viscous flow was reached, regardless the concentration of CD.

A slight increase in G* was observed in the suspensions reinforced with higher quantities of RCDs, except for 0.25% RCDs where the increase in the G* modulus at frequencies over 10 Hz was superior to the other samples. Figure 6 (Appendix A for APCDs) shows the loss tangent (tan δ) versus frequency for the different CAS/HMP suspensions. All the samples showed to have a predominantly viscous behavior at low frequencies and elastic behavior at high frequencies. The CAS/HMP suspension containing 1% RCDs had the lowest tan delta values over the whole frequency range and also the least frequency dependency demonstrating a more complex structure than the other suspensions.

### 3.4. Dopped CAS/HMP Biopolymer Films

The CAS/HMP films containing different concentrations of gCDs, prepared from a 25:75 *v*/*v* ratio, according to the protocol described in the experimental section, showed a manageable structure and a solid appearance (Figure 7). The films with the absence of gCDs were transparent and less rigid than those containing CDs, which presented a brownish color, similar to that of the neat CD dispersions. When the percentage of CDs added to the mixtures was higher, the final coloration of the films appeared to be darker and the apparent stiffness increased. The presence of more and less dark spots in all the samples could have been caused by agglomerations of the gCDs formed during the obtention of the films. As explained in the work of Zhuang et al. [29], the process of drying casein–pectin films can gradually lead to the immobilization of the casein–pectin microparticles in the casein–pectin matrix. The formation of such particles could have prevented the homogeneous distribution of the gCDs in the medium during the drying stage. SEM images of the cross-section of all biopolymer films showed that, regardless of CD concentration, all samples exhibited a compact structure with a mean thickness of 83 ± 2 nm and the presence of micropores. Figure 8 shows two SEM representative images for a film containing 0.25% of RCDs, (CAS/HMP + 0.25 RCDs biofilm), where these morphological characteristics can be appreciated.

#### Mechanical Properties

The analysis by dynamic mechanical test allows for the study of the mechanical properties of biopolymer films in a wide range of temperatures and relative humidity, in order to explore the possible applications based on their response to the changes in different environmental conditions. A first test was performed at variable temperatures with CAS/HMP and CAS/HMP + 1.0CD films. The mechanical properties of the biopolymer films were graphically represented using the values of the complex modulus (E*) obtained after each measurement. The complex modulus is defined as the sum of the storage modulus (E’) and the loss module of energy (E”), and it is a parameter used to describe the resistance of films due to mechanical stress [30]. For each film, the ramps of temperature were obtained at three different levels of relative humidity (10%, 50% and 70%). It is visible in Figure 9 that both the CAS/HMP biofilm and the one that contained 1.0% *w*/*w* of RCDs (Figure 9b and Appendix A for APCDs) presented the highest E* value under low conditions of relative humidity (10% RH), showing good responses to stress up to 45 °C. It can also be noted that the complex modulus values of RCDs are higher than CAS–HMP (e.g., 825 mPa compared to 200 mPa of CAS–HMP at 25 °C), demonstrating a more elaborate molecular structure formed by the chemical bonds of the polymers and the high concentration of gCDs. An increase in the Young modulus (E) was also observed by Babaee et al. [31] in starch films reinforced with chitosan nanoparticles. The biopolymer films analyzed in this study tended to lose their structure (lower E* values) at temperatures above 30 °C, when the relative humidity was over 50%. However, the gCD-reinforced CAS/HMP biofilm showed better stability at 30 °C at 50% or 70% RH. Under conditions of high relative humidity (70%), both film samples were easily deformed (E* very low) and it was not possible to detect a mechanical response above 31 °C. This can be explained because of the nature of the starting materials used for the obtention of the biopolymer films. In fact, despite the final solubility, the wetting and soluble properties of pectin, the component present in greater quantities in the analyzed biofilms, are often scarce. It has been observed that the conditions of high temperature and humidity caused not only molecular chemical alterations, increasing the hydrophilic groups by demethoxylation and depolymerization, but also greatly affected pectin–water interactions. The influence of environmental parameters on the molecular structure of pectin in addition to the action of the hydrophilic nature of the carbon dots obtained through hydrothermal synthesis [32] can be the main reasons that lead to the high susceptibility of the films to water absorption and consequently to the loss of their structure.

In order to examine the resistance of the films to tension stress for a longer period of time, a second mechanical test was performed for CAS/HMP films, and the gCDs reinforced the biofilm at four different temperatures and 50% relative humidity. Figure 10a shows that, at 18 °C, the CAS/HMP films have stable E* values throughout the testing time and that CAS/HMP films containing RCDs show higher E* values. Films with a high RCD content (1.0% *w*/*w*) showed higher E* values than the rest, which implies that by increasing the number of nanoparticles in the mixture, the interactions between the components are promoted causing a greater rigidity of the structure of the biopolymer films [23]. In the test realized at 25 °C (Figure 10b), a great loss of structural strength (E* decreases notably) was observed in all films, particularly in those reinforced with 0.5% and 1.0% *w*/*w* RCDs. It is interesting to note that, at this temperature, biopolymer films with a lower content of RCDs were slightly more stable than the rest. Despite the loss of strength of the structure, the measurement of all the samples was possible for the duration of the test. The responses to stress changed remarkably in the tests performed at 30 and 35 °C, (Figure 10c,d), where it can be observed that only the CAS/HMP film and the samples with a lower percentage of gCDs (0.25% *w*/*w*) maintain a certain structural stability. Thus, at 35 °C, both CAS/HMP and CAS/HMP + 0.25RCD biopolymer films retained at least 30% of the stability observed at 18 °C. Therefore, all films are subject to the diffusion of water present in the atmosphere through micropores (see Figure 8), which causes a loss of resistance in the structure, although films with a higher concentration of gCDs (0.5% and 1.0% *w*/*w*) are the most susceptible to this effect at temperatures ≥ 30 °C. Similar results were obtained using RCDs and APCDs (see data for APCDs, Appendix A). In summary, we can state that the addition of nanoparticles increases the strength of the structure of CAS/HMP biocompatible films, making them stable at mild temperatures under conditions of relative humidity lower than 50%.

## 4. Conclusions

In the present study, mixtures of sodium caseinate (CAS) and highly methoxylated pectin (HMP), as well as suspensions enriched with green carbon dots (gCDs) obtained from plant biomass (the leaves of rosemary and apple pomace), were fabricated. The suspensions of the precursor films were shown to have a shear thinning behavior, and shear sensitivity was more pronounced for the samples that were reinforced with gCDs demonstrating that the reinforced molecular network can influence the rheological test results. The biopolymer films had a manageable structure and a compact appearance, and the results prove that gCDs favorably modify the thermal stability as evidenced by the changes in crystallization temperature. The novel data obtained from the mechanical studies of the films under a relative humidity below 50% at temperatures of 25 °C or lower indicated that the stability of the biofilms was improved in the presence of gCDs. This study is not only a step towards the application of green CDs in the polymer industry, but these parameters also foresee potential applications of enriched biopolymer films as protective packaging material in situations where humidity and temperature conditions may be controlled, and the above limits are not exceeded.

## Figures and Tables

**Figure 1 membranes-12-00695-f001:**
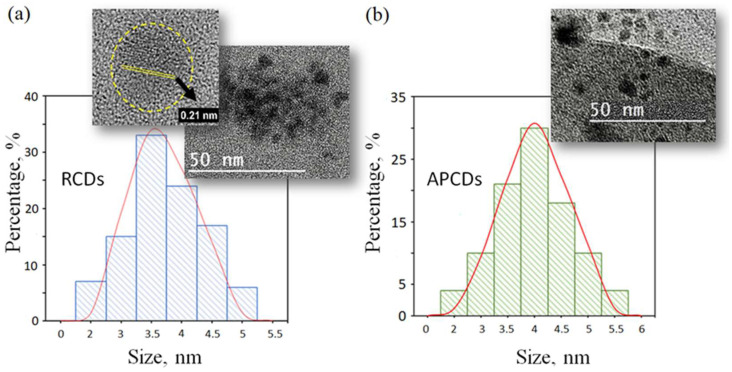
Size histograms of RCDs (**a**) and APCDs (**b**) with HR-TEM images. Magnified section: crystalline phase with an interplanar spacing of 0.21 nm.

**Figure 2 membranes-12-00695-f002:**
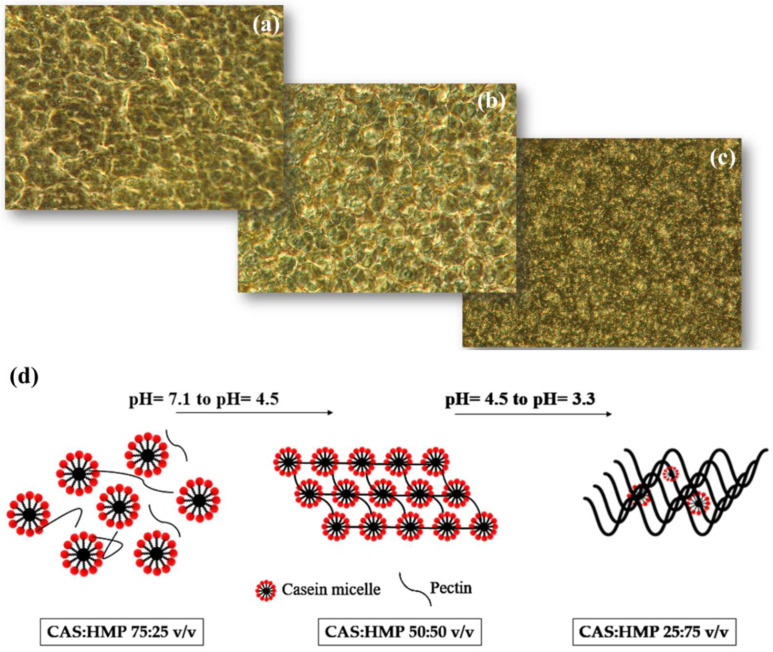
Images obtained by inverse microscope for different CAS:HMP ratio dispersions (magnification 20 × 0.40), (**a**) 75:25 *v*/*v*, (**b**) 50:50 *v*/*v* and (**c**) 25:75 *v*/*v*, and schematization of the different molecular structures of the samples (**d**).

**Figure 3 membranes-12-00695-f003:**
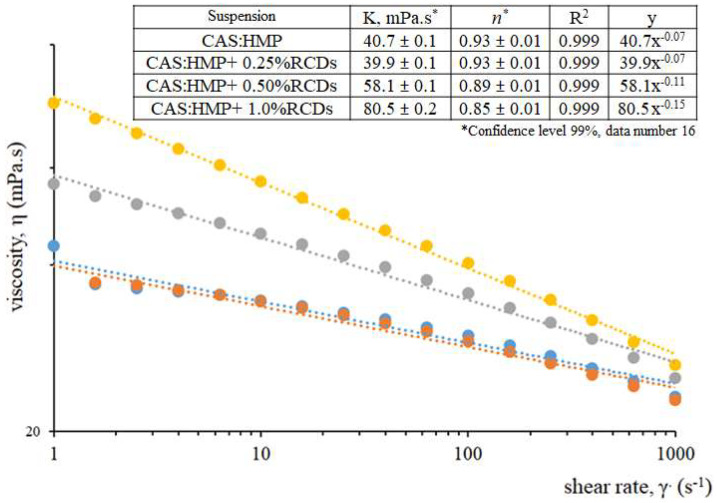
Viscosity flow curves (log–log plot) of CAS:HMP suspensions (25:75 *v*/*v* ratio, pH 3.81 ± 0.4) in the absence and presence of RCDs measured at 25 °C. Inset table: flow parameter •CAS/HMP, •CAS/HMP + 0.25%RCDs, •CAS/HMP + 0.5%RCDs, •CAS/HMP + 1.0%RCDs.

**Figure 4 membranes-12-00695-f004:**
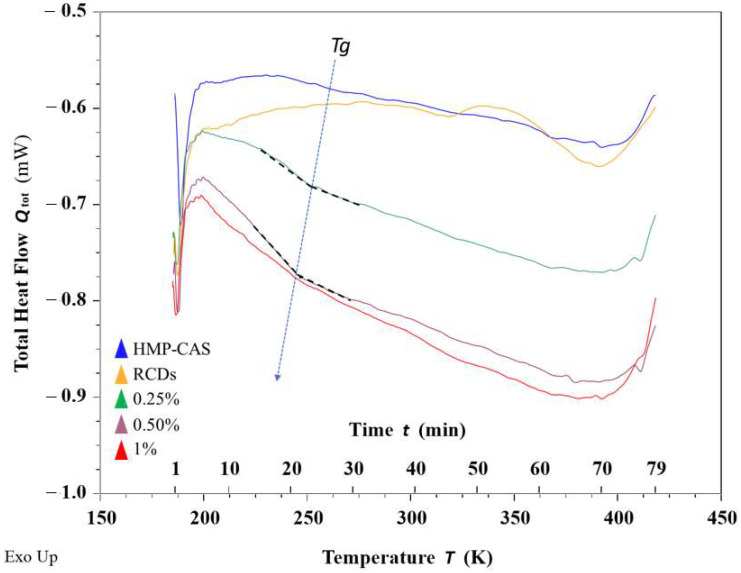
DSC thermograms of CAS/HMP films reinforced with RCDs.

**Figure 5 membranes-12-00695-f005:**
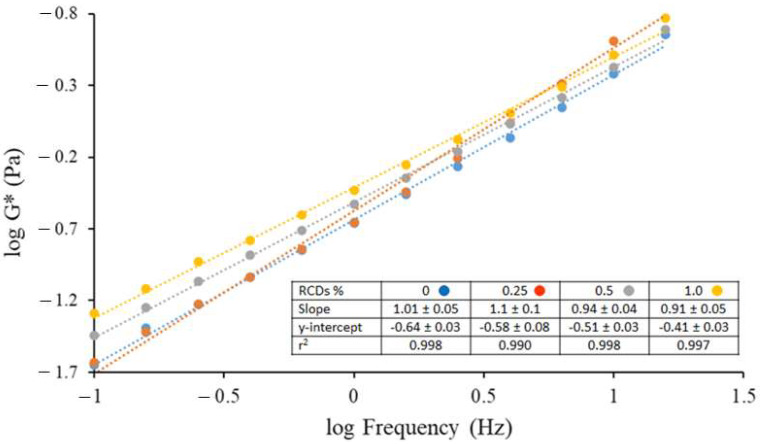
Complex modulus versus frequency in double logarithmic scales for CAS/HMP suspensions containing different amounts of RCDs: 0% (—), 0.25% (—), 0.5% (—) and 1.0% (—).

**Figure 6 membranes-12-00695-f006:**
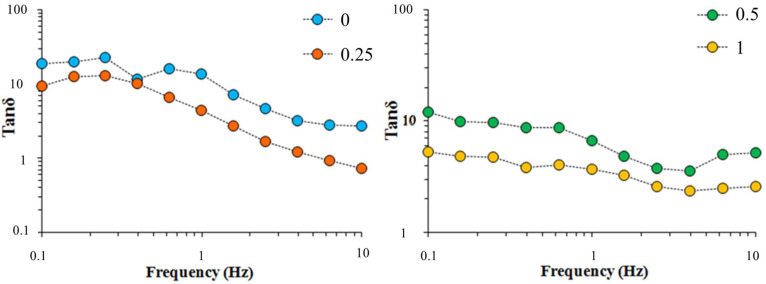
Loss tangent versus frequency for CAS/HMP suspensions containing different amounts of RCDs: 0% (—), 0.25% (—), 0.5% (—) and 1.0% (—).

**Figure 7 membranes-12-00695-f007:**
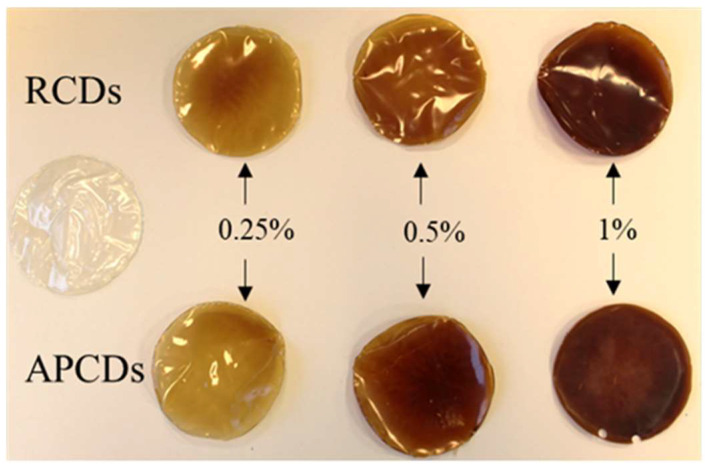
Appearance and coloration of CAS/HMP (25:75 *v*/*v* ratio) biopolymer films in the absence or reinforced with different concentrations of gCDs obtained from rosemary (RCD) and apple pomace (APCD).

**Figure 8 membranes-12-00695-f008:**
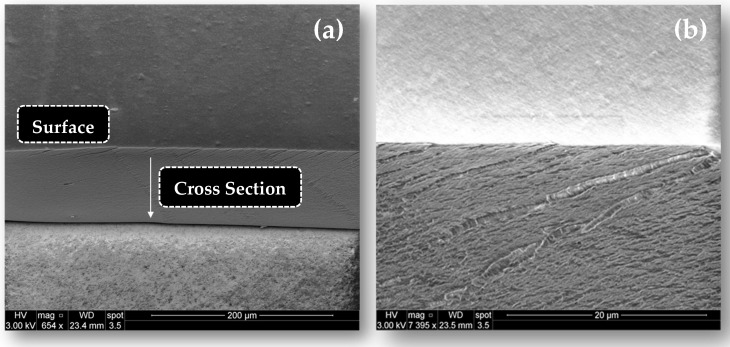
SEM images of a CAS/HMP (25:75) biofilm containing 0.25% RCDs at different magnifications: 200 μm and 20 μm.

**Figure 9 membranes-12-00695-f009:**
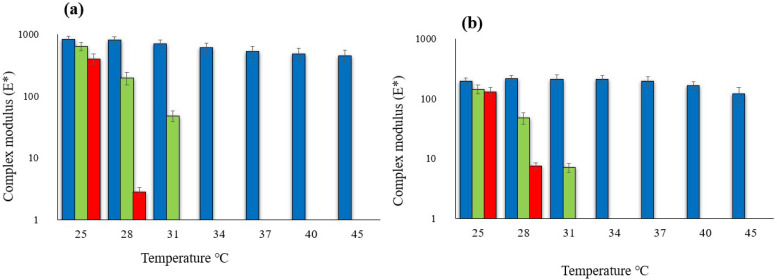
Rheological behavior as a function of temperature and relative humidity (•10% RH, •50%RH, •70%RH.) of biopolymer films: (**a**) CAS/HMP + 1.0 RCDs and (**b**) CAS/HMP.

**Figure 10 membranes-12-00695-f010:**
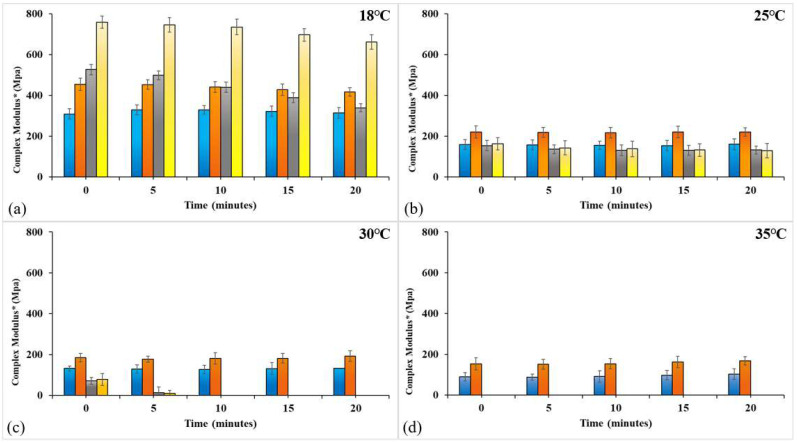
Mechanical responses of biopolymer films as a function of exposure time at different temperatures and an RH of 0,50%, • CAS/HMP, • CAS/HMP + 0.25RCDs, • CAS/HMP + 0.5RCD, • CAS/HMP + 1.0RCD.

## Data Availability

The data presented in this study are available on request from the corresponding author.

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
