# Peer review of "High Methoxyl Pectin and Sodium Caseinate Film Matrix Reinforced with Green Carbon Quantum Dots: Rheological and Mechanical Studies"

_membranes, 2022, doi:10.3390/membranes12070695_

Round 1
Reviewer 1 Report
Dear authors,
I consider your paper "High metoxyl pectin and sodium caseinate biofilms matrix reinforced with green carbon quantum dots: rheological and mechanical studies" is appropriate for being published in Membranes.
Reviewer 2 Report
The manuscript (membranes-1794000) presents a work on the development of “High metoxyl pectin and sodium caseinate biofilms matrix reinforced with green carbon quantum dots: rheological and mechanical studies”. The analysis carried out is minimal, giving a fair description of its properties. Moreover, the discussion is written poorly. A seprate results section and a separate discussion section would greatly enrich the article. Additionally, the main problem with this article is lack of statistical analysis.
General comments are as follows:
11. In the title: “Methoxyl”, not “metoxyl”.
22. I found some of the writing difficult to follow. I suggest a careful edit for grammar and readability.
33. Novelty and rationale behind the research need to be mentioned in more detail in introduction section.
44. Figure 2: Scale bar for the images should be mentioned and better-quality images should be provided.
55. Section 3.4: “from a 25:75 v/v ratio, showed a homogeneous and flexible appearance (Figure 7).” The provided images and claims do not match. There are dark and light spots are present in every film so I doubt the homogeneity. Moreover, the use of flexible appearance seems quite awkward.
66. The results and discussion section is written poorly. Authors must elaborate this section and provide elaborated discussion rather than just describing the results. Authors are encouraged to compare the obtained results with current research and provide scientific insights based on the results obtained.
77. Statistical analysis must be performed.
88. Conclusion could be more attractive if authors include key insights and future prospective instead of just summarizing the results.
99. Authors have not cited relevant research in this manuscript, many statements/claims need proper reference/citation.
110. References present some formatting issues, please check page numbers and other details carefully for book chapters.
Reviewer 3 Report
Dear Authors,
The submitted manuscript dealing with pectin/caseinate based films enriched with green carbon dots provides the valuable results with potential for food packaging applications. However, the text suffers from a number of shortages that have to be considered before further processing in Membranes Journal, which are listed below:
- I do not find appropriate the use of term „biofilm“ in the context of given manuscript. “Biofilm” defines the clusters of bacteria attached to the surface while in the submitted manuscript the term describes the polymer film prepared by solvent casting from biopolymers. It should be replaced throughout the whole text, e.g. with either “film”, or “biopolymer film”.
- Specification of DMA measuring equipment (supplier, country) should be added in Chapter 2.6.
- More detailed specification of DSC measurement should be given (from what scan the data in Figure 4 are shown?).
- In Fig. S2 the indication of Tg is missing in the thermogram. The Figure titles in Supplementary (Fig. S2, S3 and S4) has to be edited (APCDs instead of RCDs).
- The explanation of abbreviation should be included when first mentioned (HRTEM, line 183)
- In line 199, Figure 2a (instead to Figure 1a) should be referred.
- Title of Figure 2 should be shifted below the Figure (line 220).
- Equation 1 in line 241 should be located in the centre of the page.
- In line 244, the gap between Figure3 should be inserted (to Figure 3).
- In line 249 “polymers chains” should be replaced with “polymer chains”.
- The DSC results should be given in unified units in Figure 4 and discussion (in Chapter 3.2.), probably in °C will be more appropriate.
- Decimal points (dots vs. commas) should be unified in the text (e.g. legend in Figure 6 contains commas while decimal dots are mentioned in Figure title).
- Line 357 – units of mean thickness has to be edited.
- The quality of Fig. 10 should be improved, column graphs with light coloured boundaries are not clearly visible. The relative humidity of 50 % is mentioned in the text (line 412), while in Figure 10 title, the RH 0 % is given. This must be unified.
- The authors’ names should be added in Ref. 22 and Ref. 24.
Reviewer 4 Report
Dear Authors
Your article is well organized. It is a novel study that combines biomolecules with CDs intended for food packaging. Interesting story that introduce CDs in packaging materials. There are limited number of such articles, so this article will be of a big interest for readership. I recommend this article for publishing after consideration/addressing the suggestions/comments that are given in the attached document.
Thank you.

Round 2
Reviewer 2 Report
The authors have addressed all comments satisfactorily. I now consider this manuscript suitable for publication.